# Theoretical evidence for adversarial robustness through randomization

**Rafael Pinot**[1,2]   **Laurent Meunier**[1,3]   **Alexandre Araujo**[1,4]
**Hisashi Kashima**[5,6]   **Florian Yger**[1]   **Cédric Gouy-Pailler**[2]   **Jamal Atif**[1]

[1]Université Paris-Dauphine, PSL Research University, CNRS, LAMSADE, Paris, France
[2]Institut LIST, CEA, Université Paris-Saclay    [3]Facebook AI Research, Paris, France
[4]Wavestone, Paris, France    [5]Kyoto University, Kyoto, Japan    [6]RIKEN Center for AIP, Japan

## Abstract

This paper investigates the theory of robustness against adversarial attacks. It focuses on the family of randomization techniques that consist in injecting noise in the network at inference time. These techniques have proven effective in many contexts, but lack theoretical arguments. We close this gap by presenting a theoretical analysis of these approaches, hence explaining why they perform well in practice. More precisely, we make two new contributions. The first one relates the randomization rate to robustness to adversarial attacks. This result applies for the general family of exponential distributions, and thus extends and unifies the previous approaches. The second contribution consists in devising a new upper bound on the adversarial risk gap of randomized neural networks. We support our theoretical claims with a set of experiments.

## 1   Introduction

Adversarial attacks are some of the most puzzling and burning issues in modern machine learning. An adversarial attack refers to a small, imperceptible change of an input maliciously designed to fool the result of a machine learning algorithm. Since the seminal work of [42] exhibiting this intriguing phenomenon in the context of deep learning, a wealth of results have been published on designing attacks [18, 34, 32, 23, 6, 31] and defenses [18, 35, 20, 29, 39, 27]), or on trying to understand the very nature of this phenomenon [17, 40, 15, 16]. Most methods remain unsuccessful to defend against powerful adversaries [6, 28, 1]. Among the defense strategies, randomization has proven effective in some contexts. It consists in injecting random noise (both during training and inference phases) inside the network architecture, *i.e.* at a given layer of the network. Noise can be drawn either from Gaussian [26, 24, 37], Laplace [24], Uniform [44], or Multinomial [12] distributions. Remarkably, most of the considered distributions belong to the Exponential family. Albeit these significant efforts, several theoretical questions remain unanswered. Among these, we tackle the following, for which we provide principled and theoretically-founded answers:

**Q1:** To what extent does a noise drawn from the Exponential family preserve robustness (in a sense to be defined) to adversarial attacks?

**A1:** We introduce a definition of robustness to adversarial attacks that is suitable to the randomization defense mechanism. As this mechanism can be described as a non-deterministic querying process, called probabilistic mapping in the sequel, we propose a formal definition of robustness relying on a metric/divergence between probability measures. A key question arises then about the appropriate metric/divergence for our context. This requires tools for comparing divergences w.r.t. the introduced robustness definition. Renyi divergence turned out to be a measure of choice, since it satisfies most of

the desired properties (coherence, strength, and computational tractability). Finally, thanks to the existing links between Renyi divergence and the Exponential family, we prove that methods based on noise injection from the Exponential family ensures robustness to adversarial attacks (cf Theorem 1).

**Q2:** Can we guarantee a good accuracy under attack for classifiers defended with these noises?

**A2:** We present an upper bound on the drop of accuracy (under attack) of the methods defended with noise drawn from the Exponential family (cf. Theorem 2). We also provide a certificate on accuracy under attack for this kind of noise (cf Theorem 3). We illustrate this result by training different randomized models with Laplace and Gaussian distributions on CIFAR10/CIFAR100. These experiments highlight the trade-off between accuracy and robustness that depends on the amount of noise one injects in the network. Our theoretical and experimental conclusion is that randomized defenses are competitive (with the state-of-the-art [28]) given the intensity of the injected noise.

**Outline of the paper:** We present in Section 2 the related work on randomized defenses to adversarial examples. Section 3 introduces the definition of robustness relying on a metric/divergence between probability measures, and discusses the key role of the Renyi divergence. We state in Section 4 our main results on the robustness and accuracy of Exponential family-based defenses. Section 5 presents extensive experiments supporting our theoretical findings. Section 6 provides concluding remarks.

## 2 Related works

Injecting noise into algorithms to improve their robustness has been used for ages in detection and signal processing tasks [46, 7, 30]. It has also been extensively studied in several machine learning and optimization fields, *e.g.* robust optimization [4] and data augmentation techniques [36]. Recently, noise injection techniques have been adopted by the adversarial defense community, especially for neural networks, with very promising results. Randomization techniques are generally oriented towards one of the following objectives: experimental robustness or provable robustness.

**Experimental robustness:** The first technique explicitly using randomization at inference time as a defense appeared during the 2017 NIPS defense challenge [44]. This method uniformly samples over geometric transformations of the image to select a substitute image to feed the network. Then [12] proposed to use stochastic activation pruning based on a multinomial distribution for adversarial defense. Several papers [26, 37] propose to inject Gaussian noise directly on the activation of selected layers both at training and inference time. While these works hypothesize that noise injection makes the network robust to adversarial perturbations, they do not provide any formal justification on the nature of the noise they use or on the loss of accuracy/robustness of the network.

**Provable robustness:** In [24], the authors proposed a randomization method by exploiting the link between differential privacy [14] and adversarial robustness. Their framework, called "randomized smoothing" [1], inherits some theoretical results from the differential privacy community allowing them to evaluate the level of accuracy under attack of their method. Initial results from [24] have been refined in [25], and [9]. Our work belongs to this line of research. However, our framework does not treat exactly the same class of defenses. Notably, we provide theoretical arguments supporting the defense strategy based on randomization techniques relying on the exponential family, and derive a new bound on the adversarial risk gap, which completes the results obtained so far on certified robustness. Furthermore, our main focus is on the network randomized by noise injection, "randomized smoothing" instead uses this network to create a *new* classifier robust to attacks.

Since the initial discovery of adversarial examples, a wealth of non randomized defense approaches have also been proposed, inspired by various machine learning domains such as adversarial training [18, 27], image reconstruction [29, 39] or robust learning [18, 27]. Even if these methods have their own merits, a thorough evaluation made by [1] shows that most defenses can be easily broken with known powerful attacks [27, 6, 8]. Adversarial training, which consists in training a model directly on adversarial examples, came out as the best defense in average. Defense based on randomization could be overcome by the Expectation Over Transformation technique proposed by [2] which consists in taking the expectation over the network to craft the perturbation. In this paper, to ensure that our results are not biased by obfuscated gradients, we follow the principles of [1, 5] and evaluate our randomized networks with this technique. We show that randomized defenses are still competitive given the intensity of noise injected in the network.

# 3 General definitions of risk and robustness

## 3.1 Risk, robustness and probabilistic mappings

Let us consider two spaces $\mathcal{X}$ (with norm $\|.\|_{\mathcal{X}}$), and $\mathcal{Y}$. We consider the classification task that seeks a hypothesis (classifier) $h : \mathcal{X} \rightarrow \mathcal{Y}$ minimizing the risk of $h$ w.r.t. some ground-truth distribution $\mathcal{D}$ over $\mathcal{X} \times \mathcal{Y}$. The risk of $h$ w.r.t $\mathcal{D}$ is defined as $\text{Risk}(h) := \mathbb{E}_{(x,y)\sim\mathcal{D}} [\mathbb{1}\, (h(x) \neq y)]$. Given a classifier $h : \mathcal{X} \rightarrow \mathcal{Y}$, and some input $x \in \mathcal{X}$ with true label $y_{true} \in \mathcal{Y}$, to generate an adversarial example, the adversary seeks a $\tau$ such that $h(x + \tau) \neq y_{true}$, with some budget $\alpha$ over the perturbation (*i.e* with $\|\tau\|_{\mathcal{X}} \leq \alpha$). $\alpha$ represents the maximum amount of perturbation one can add to $x$ without being spotted (the perturbation remains humanly imperceptible). The overall goal of the adversary is to find a perturbation crafting strategy that both maximizes the risk of $h$, and keeps the values of $\|\tau\|_{\mathcal{X}}$ small. To measure this risk "under attack" we define the notion of adversarial $\alpha$-radius risk of $h$ w.r.t. $\mathcal{D}$ as follows

$$\text{Risk}_\alpha(h) := \mathbb{E}_{(x,y)\sim\mathcal{D}} \left[ \sup_{\|\tau\|_{\mathcal{X}} \leq \alpha} \mathbb{1}\, (h(x + \tau) \neq y) \right] \ .$$

In practice, the adversary does not have any access to the ground-truth distribution. The literature proposed several surrogate versions of $\text{Risk}_\alpha(h)$ (see [13] for more details) to overcome this issue. We focus our analysis on the one used in *e.g* [42], or [15] denoted $\alpha$-radius prediction-change risk of $h$ w.r.t. $\mathcal{D}_{\mathcal{X}}$ (marginal of $\mathcal{D}$ for $\mathcal{X}$), and defined as

$$\text{PC-Risk}_\alpha(h) := \mathbb{P}_{x\sim\mathcal{D}_{\mathcal{X}}} [\exists \tau \in \text{B}(\alpha) \text{ s.t. } h(x + \tau) \neq h(x)]$$

where for any $\alpha \geq 0$, $\text{B}(\alpha) := \{\tau \in \mathcal{X} \text{ s.t. } \|\tau\|_{\mathcal{X}} \leq \alpha\}$ .

As we will inject some noise in our classifier in order to defend against adversarial attacks, we need to introduce the notion of "probabilistic mapping". Let $\mathcal{Y}$ be the output space, and $\mathcal{F}_{\mathcal{Y}}$ a $\sigma\text{-}algebra$ over $\mathcal{Y}$. Let us also denote $\mathcal{P}(\mathcal{Y})$ the set of probability measures over $(\mathcal{Y}, \mathcal{F}_{\mathcal{Y}})$.

**Definition 1** (Probabilistic mapping). *Let $\mathcal{X}$ be an arbitrary space, and $(\mathcal{Y}, \mathcal{F}_{\mathcal{Y}})$ a measurable space. A* probabilistic mapping *from $\mathcal{X}$ to $\mathcal{Y}$ is a mapping $\text{M} : \mathcal{X} \rightarrow \mathcal{P}(\mathcal{Y})$. To obtain a numerical output out of this* probabilistic mapping*, one needs to sample $y$ according to $\text{M}(x)$.*

This definition does not depend on the nature of $\mathcal{Y}$ as long as $(\mathcal{Y}, \mathcal{F}_{\mathcal{Y}})$ is measurable. In that sense, $\mathcal{Y}$ could be either the label space or any intermediate space corresponding to the output of an arbitrary hidden layer of a neural network. Moreover, any mapping can be considered as a probabilistic mapping, whether it explicitly injects noise (as in [24, 37, 12]) or not. In fact, any deterministic mapping can be considered as a probabilistic mapping, since it can be characterized by a Dirac measure. Accordingly, the definition of a probabilistic mapping is fully general and equally treats networks with or without noise injection. There exists no definition of robustness against adversarial attacks that comply with the notion of probabilistic mappings. We settle that by generalizing the notion of prediction-change risk initially introduced in [13] for deterministic classifiers. Let $\text{M}$ be a probabilistic mapping from $\mathcal{X}$ to $\mathcal{Y}$, and $d_{\mathcal{P}(\mathcal{Y})}$ some metric/divergence on $\mathcal{P}(\mathcal{Y})$. We define the $(\alpha, \epsilon)$-radius prediction-change risk of $\text{M}$ w.r.t. $\mathcal{D}_{\mathcal{X}}$ and $d_{\mathcal{P}(\mathcal{Y})}$ as

$$\text{PC-Risk}_\alpha(\text{M}, \epsilon) := \mathbb{P}_{x\sim\mathcal{D}_{\mathcal{X}}} \left[ \exists \tau \in B(\alpha) \text{ s.t. } d_{\mathcal{P}(\mathcal{Y})}(\text{M}(x + \tau), \text{M}(x)) > \epsilon \right] \ .$$

These three generalized notions allow us to analyze noise injection defense mechanisms (Theorems 1, and 2). We can also define adversarial robustness (and later adversarial gap) thanks to these notions.

**Definition 2** (Adversarial robustness). *Let $d_{\mathcal{P}(\mathcal{Y})}$ be a metric/divergence on $\mathcal{P}(\mathcal{Y})$. A probabilistic mapping $\text{M}$ is called $d_{\mathcal{P}(\mathcal{Y})}$-$(\alpha, \epsilon, \gamma)$ robust if $\text{PC-Risk}_\alpha(\text{M}, \epsilon) \leq \gamma$, $d_{\mathcal{P}(\mathcal{Y})}$-$(\alpha, \epsilon)$ robust if $\gamma = 0$.*

It is difficult in general to show that a classifier is $d_{\mathcal{P}(\mathcal{Y})}$-$(\alpha, \epsilon, \gamma)$ robust. However, we can derive some bounds for particular divergences that will ensure robustness up to a certain level (Theorem 1). It is worth noting that our definition of robustness depends on the considered metric/divergence between probability measures. Lemma 1 gives some insights on the monotony of the robustness according to the parameters, and the probability metric/divergence at hand.

**Lemma 1.** *Let $\text{M}$ be a probabilistic mapping, and let $d_1$ and $d_2$ be two metrics on $\mathcal{P}(\mathcal{Y})$. If there exists a non decreasing function $\phi : \mathbb{R} \rightarrow \mathbb{R}$ such that $\forall \mu_1, \mu_2 \in \mathcal{P}(\mathcal{Y})$, $d_1(\mu_1, \mu_2) \leq \phi(d_2(\mu_1, \mu_2))$, then the following assertion holds: $\text{M}$ is $d_2$-$(\alpha, \epsilon, \gamma)$-robust $\implies$ $\text{M}$ is $d_1$-$(\alpha, \phi(\epsilon), \gamma)$-robust.*

As suggested in Definition 2 and Lemma 1, any given choice of metric/divergence will instantiate a particular notion of adversarial robustness and it should be carefully selected.

## 3.2 On the choice of the metric/divergence for robustness

The aforementioned formulation naturally raises the question of the choice of the metric used to defend against adversarial attacks. The main notions that govern the selection of an appropriate metric/divergence are *coherence*, *strength*, and *computational tractability*. A metric/divergence is said to be coherent if it naturally fits the task at hand (*e.g.* classification tasks are intrinsically linked to discrete/trivial metrics, conversely to regression tasks). The strength of a metric/divergence refers to its ability to cover (dominate) a wide class of others in the sense of Lemma 1. In the following, we will focus on both the total variation metric and the Renyi divergence, that we consider as respectively the most coherent with the classification task using probabilistic mappings, and the strongest divergence. We first discuss how total variation metric is *coherent* with randomized classifiers but suffers from computational issues. The Renyi divergence provides good guarantees about adversarial robustness, enjoys nice *computational properties*, in particular when considering Exponential family distributions, and is *strong* enough to dominate a wide range of metrics/divergences including total variation.

Let $\mu_1$ and $\mu_2$ be two measures in $\mathcal{P}(\mathcal{Y})$, both dominated by a third measure $\nu$. The trivial distance $d_T(\mu_1, \mu_2) := \mathbb{1}(\mu_1 \neq \mu_2)$ is the simplest distance one can define between $\mu_1$ and $\mu_2$. In the deterministic case, it is straightforward to compute (since the numerical output of the algorithm characterizes its associated measure), but this is not the case in general. In fact one might not have access to the true distribution of the mapping, but just to the numerical outputs. Therefore, one needs to consider more sophisticated metrics/divergences, such as the total variation distance $d_{TV}(\mu_1, \mu_2) := \sup_{Y \in \mathcal{F}_{\mathcal{Y}}} |\mu_1(Y) - \mu_2(Y)|$. The total variation distance is one of the most broadly used probability metrics. It admits several very simple interpretations, and is a very useful tool in many mathematical fields such as probability theory, Bayesian statistics, coupling or transportation theory. In transportation theory, it can be rewritten as the solution of the Monge-Kantorovich problem with the cost function $c(y_1, y_2) = \mathbb{1}(y_1 \neq y_2)$: $\inf \int_{\mathcal{Y}^2} \mathbb{1}(y_1 \neq y_2) \, d\pi(y_1, y_2)$, where the infimum is taken over all joint probability measures $\pi$ on $(\mathcal{Y} \times \mathcal{Y}, \mathcal{F}_{\mathcal{Y}} \otimes \mathcal{F}_{\mathcal{Y}})$ with marginals $\mu_1$ and $\mu_2$. According to this interpretation, it seems quite natural to consider the total variation distance as a relaxation of the trivial distance on $[0, 1]$ (see [43] for details). In the deterministic case, the total variation and the trivial distance coincides. In general, the total variation allows a finer analysis of the probabilistic mappings than the trivial distance. But it suffers from a high computational complexity. In the following of the paper we will show how to ensure robustness regarding TV distance.

Finally, denoting by $g_1$ and $g_2$ the respective probability distributions w.r.t. $\nu$, the Renyi divergence of order $\lambda$ [38] writes as $d_{R,\lambda}(\mu_1, \mu_2) := \frac{1}{\lambda - 1} \log \int_{\mathcal{Y}} g_2(y) \left( \frac{g_1(y)}{g_2(y)} \right)^\lambda d\nu(y)$. The Renyi divergence is a generalized measure defined on the interval $(1, \infty)$, where it equals the Kullback-Leibler divergence when $\lambda \to 1$ (that will be denoted $d_{KL}$), and the maximum divergence when $\lambda \to \infty$. It also has the very special property of being non decreasing w.r.t. $\lambda$. This divergence is very common in machine learning, especially in its Kullback-Leibler form as it is widely used as the loss function (cross entropy) of classification algorithms. It enjoys the desired properties since it bounds the TV distance, and is tractable. Furthermore, Proposition 1 proves that Renyi-robustness implies TV-robustness, making it a suitable surrogate for the trivial distance.

**Proposition 1** (Renyi implies TV-robustness)**.** *Let* M *be a probabilistic mapping, then for all* $\lambda \geq 1$, $\epsilon > 0$, *there exists* $\epsilon' > 0$ *s.t. if* M *is* $d_{R,\lambda}$-$(\alpha, \epsilon, \gamma)$-robust then M *is* $d_{TV}$-$(\alpha, \epsilon', \gamma)$-robust.

A crucial property of Renyi-robustness is the *Data processing inequality*. It is a well-known inequality from information theory which states that *"post-processing cannot increase information"* [10, 3]. In our case, if we consider a Renyi-robust probabilistic mapping, composing it with a deterministic mapping maintains Renyi-robustness with the same level.

**Proposition 2** (Data processing inequality)**.** *Let us consider a probabilistic mapping* M : $\mathcal{X} \to \mathcal{P}(\mathcal{Y})$, *and denote* $\rho : \mathcal{Y} \to \mathcal{Y}'$ *a deterministic function. If* $U \sim \mathrm{M}(x)$ *then the probability measure* $M'(x)$ *s.t.* $\rho(U) \sim M'(x)$ *defines a probabilistic mapping* $M' : \mathcal{X} \to \mathcal{P}(\mathcal{Y}')$. *For any* $\lambda > 1$, *if* M *is* $d_{R,\lambda}$-$(\alpha, \epsilon, \gamma)$ *robust then* $M'$ *is also* $d_{R,\lambda}$-$(\alpha, \epsilon, \gamma)$ *robust.*

Data processing inequality will allow us later to inject some additive noise in any layer of a neural network and to ensure Renyi-robustness.

# 4 Defense mechanisms based on Exponential family noise injection

## 4.1 Robustness through Exponential family noise injection

For now, the question of which class of noise to add is treated *ad hoc*. We choose here to investigate one particular class of noise closely linked to the Renyi divergence, namely Exponential family distributions, and demonstrate their interest. Let us first recall what the Exponential family is.

**Definition 3** (Exponential family). *Let $\Theta$ be an open convex set of $\mathbb{R}^n$, and $\theta \in \Theta$. Let $\nu$ be a measure dominated by $\mu$ (either by the Lebesgue or counting measure), it is said to be part of the Exponential family of parameter $\theta$ (denoted $E_F(\theta, t, k)$) if it has the following p.d.f.*

$$p_F(z, \theta) = \exp\left\{\langle t(z), \theta \rangle - u(\theta) + k(z)\right\}$$

*where $t(z)$ is a sufficient statistic, $k$ a carrier measure (either for a Lebesgue or a counting measure) and $u(\theta) = \log \int_z \exp\left\{< t(z), \theta > + k(z)\right\} dz$.*

To show the robustness of randomized networks with noise injected from the Exponential family, one needs to define the notion of sensitivity for a given deterministic function:

**Definition 4** (Sensitivity of a function). *For any $\alpha \geq 0$ and for any $||.||_A$ and $||.||_B$ two norms, the $\alpha$-sensitivity of $f$ w.r.t. $||.||_A$ and $||.||_B$ is defined as*

$$\Delta_\alpha^{A,B}(f) := \sup_{x,y \in \mathcal{X}, ||x-y||_A \leq \alpha} ||f(x) - f(y)||_B \;.$$

Let us consider an $n$-layer feedforward neural network $\mathcal{N}(.) = \phi^n \circ ... \circ \phi^1(.)$. For any $i \in [n]$, we define $\mathcal{N}_{|i}(.) = \phi^i \circ ... \circ \phi^1(.)$ the neural network truncated at layer $i$. Theorem 1 shows that, injecting noise drawn from an Exponential family distribution ensures robustness to adversarial example attacks in the sense of Definition 2.

**Theorem 1** (Exponential family ensures robustness). *Let us denote $\mathcal{N}_X^i(.) = \phi^n \circ ... \circ \phi^{i+1}(\mathcal{N}_{|i}(.) + X)$ with $X$ a random variable. Let us also consider two arbitrary norms $||.||_A$ and $||.||_B$ respectively on $\mathcal{X}$ and on the output space of $\mathcal{N}_X^i$.*

- *If $X \sim E_F(\theta, t, k)$ where $t$ and $k$ have non-decreasing modulus of continuity $\omega_t$ and $\omega_k$. Then for any $\alpha \geq 0$, $\mathcal{N}_X^i(.)$ defines a probabilistic mapping that is $d_{R,\lambda}$-$(\alpha, \epsilon)$ robust with $\epsilon = ||\theta||_2 \omega_t^{B,2}(\Delta_\alpha^{A,B}(\phi)) + \omega_k^{B,1}(\Delta_\alpha^{A,B}(\phi))$ where $||.||_2$ is the norm corresponding to the scalar product in the definition of the exponential family density function and $||.||_1$ is the absolute value on $\mathbb{R}$. Notions of continuity modulus is defined in the supplementary material.*

- *If $X$ is a centered Gaussian random variable with a non degenerated matrix parameter $\Sigma$. Then for any $\alpha \geq 0$, $\mathcal{N}_X^i(.)$ defines a probabilistic mapping that is $d_{R,\lambda}$-$(\alpha, \epsilon)$ robust with $\epsilon = \frac{\lambda \Delta_\alpha^{A,2}(\phi)^2}{2\sigma_{min}(\Sigma)}$ where $||.||_2$ is the canonical Euclidean norm on $\mathbb{R}^n$.*

In simpler words, the previous theorem ensures stability in the neural network when injecting noise w.r.t. the distribution of the output. Intuitively, if two inputs are close w.r.t. $||.||_A$, the output distributions of the network will be close in the sense of Renyi divergence. It is well known that in the case of deterministic neural networks, the Lipschitz constant becomes bigger as the number of layers increases [19]. By injecting noise at layer $i$, the notion of robustness only depends on the sensitivity of the first $i$ layers of the network and not the following ones. In that sense, randomization provides a more precise control on the "continuity" of the neural network. In the next section, we show that thanks to the notion of robustness w.r.t. probabilistic mappings, one can bound the loss of accuracy of a randomized neural network when it is attacked.

## 4.2 Bound on the risk gap under attack and certified accuracy

The notions of risk and adversarial risk can easily be generalized to encompass probabilistic mappings.

**Definition 5** (Risks for probabilistic mappings). *Let $M$ be a probabilistic mapping from $\mathcal{X}$ to $\mathcal{Y}$, the risk and the $\alpha$-radius adversarial risk of $M$ w.r.t. $\mathcal{D}$ are defined as envoenvoie le*

$$\text{Risk}(M) := \mathbb{E}_{(x,y) \sim \mathcal{D}} \left[ \mathbb{E}_{y' \sim M(x)} \left[ \mathbb{1}\left(y' \neq y\right) \right] \right]$$

$$\text{Risk}_\alpha(M) := \mathbb{E}_{(x,y) \sim \mathcal{D}} \left[ \sup_{||\tau||_\mathcal{X} \leq \alpha} \mathbb{E}_{y' \sim M(x+\tau)} \left[ \mathbb{1}\left(y' \neq y\right) \right] \right] \;.$$

The definition of adversarial risk for a probabilistic mapping can be matched with the concept of Expectation over Transformation (EoT) attacks [1]. Indeed, EoT attacks aim at computing the best opponent in expectation for a given random transformation. In the adversarial risk definition, the adversary chooses the perturbation which has the greatest probability to fool the model, which is a stronger objective than the EoT objective. Theorem 2 provides a bound on the gap between the adversarial risk and the regular risk:

**Theorem 2** (Adversarial risk gap bound in the randomized setting)**.** *Let* M *be the probabilistic mapping at hand. Let us suppose that* M *is* $d_{R,\lambda}$-$(\alpha, \epsilon)$ *robust for some* $\lambda \geq 1$ *then:*

$$| \operatorname{Risk}_\alpha(\mathrm{M}) - \operatorname{Risk}(\mathrm{M})| \leq 1 - e^{-\epsilon} \mathbb{E}_x \left[ e^{-H(\mathrm{M}(x))} \right]$$

*where* $H$ *is the Shannon entropy* $H(p) = -\sum_i p_i \log(p_i)$ *.*

This theorem gives a control on the loss of accuracy under attack w.r.t. the robustness parameter $\epsilon$ and the entropy of the predictor. It provides a tradeoff between the quantity of noise added in the network and the accuracy under attack. Intuitively, when the noise increases, for any input, the output distribution tends towards the uniform distribution, then, $\epsilon \to 0$ and $H(\mathrm{M}(x)) \to \log(K)$, and the risk and the adversarial risk both tends to $\frac{1}{K}$ where $K$ is the number of classes in the classification problem. On the opposite, if no noise is injected, for any input, the output distribution is a Dirac distribution, then, if the prediction for the adversarial example is not the same as for the regular one, $\epsilon \to \infty$ and $H(\mathrm{M}(x)) \to 0$. Hence, the noise needs to be designed both to preserve accuracy and robustness to adversarial attacks. In the Section 5, we give an illustration of this bound when M is a neural network with noise injection at input level as presented in Theorem 1. In practice, we do not have access to the real value of the entropy, but we estimate it with classical estimators [33].

Our framework being general enough it encompasses several known accuracy certificates from the literature, e.g. the one provided in [24]. Interestingly, we can introduce the following one, based on *our* definition of robustness.

**Theorem 3.** *Let* $x \in \mathcal{X}$, *and* M *be a probabilistic mapping with values in* $\mathbb{R}^K$. *If* M *is* $d_{R,\lambda}$-$(\alpha, \epsilon)$ *robust, and if there exist* $k^*$ *and* $\delta^* \in (0, 1)$ *s.t.* $\mathbb{E}_{y \sim \mathrm{M}(x)}[y_{k^*}] > e^{2\epsilon'} \max_{i \neq k^*} \mathbb{E}_{y \sim \mathrm{M}(x)}[y_i] + (1 + e^{\epsilon'})\delta^*,$

*with* $\epsilon' = \epsilon + \frac{\log(1/\delta^*)}{\lambda - 1}$. *Then, for the classifier* $f : x \mapsto \underset{k \in [K]}{\operatorname{argmax}} \mathbb{E}_{y \sim \mathrm{M}(x)}[y_k]$ *there is no perturbation* $\tau \in \mathrm{B}(\alpha)$ *such that* $f(x) \neq f(x + \tau)$.

As the main focus of this work is to give theoretical evidence for randomization techniques, numerical experiments will mainly focus on Theorem 1 and 2 and not on certificates (Theorem 3).

## 5 Numerical experiments

To illustrate our theoretical findings, we train randomized neural networks with a simple method which consists in injecting a noise drawn from an Exponential family distribution in the image during training and inference. This section aims to answer **Q2** stated in the introduction, by tackling the following sub-questions:

**Q2.1:** How does the randomization impact the accuracy of the network? And, how does the theoretical trade-off between accuracy and robustness apply in practice?

**Q2.2:** What is the accuracy under attack of randomized neural networks against powerful iterative attacks? And how does randomized neural networks compare to state-of-the-art defenses given the intensity of the injected noise?

### 5.1 Experimental setup

We present our results and analysis on CIFAR-10, CIFAR-100 [22] and ImageNet datasets [11]. For CIFAR-10 and CIFAR-100 [22], we used a Wide ResNet architecture [45] which is a variant of the ResNet model from [21]. We use 28 layers with a widen factor of 10. We train all networks for 200 epochs, a batch size of 400, dropout 0.3 and Leaky Relu activation with a slope on $\mathbb{R}^-$ of 0.1. We minimize the Cross Entropy Loss with Momentum 0.9 and use a piecewise constant learning

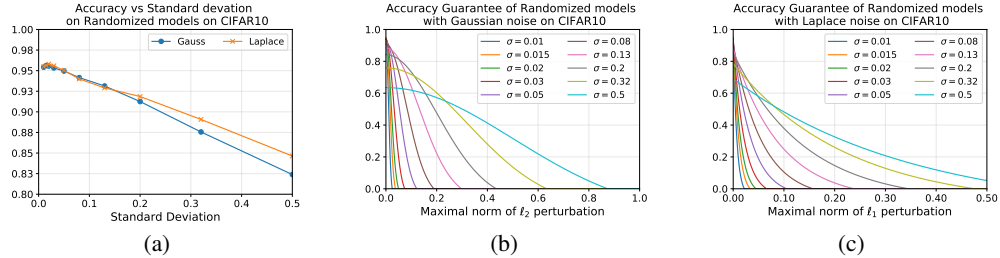

Figure 1: (a) Impact of the standard deviation of the injected noise on accuracy in a randomized model on CIFAR-10 with a Wide ResNet architecture. (b) and (c) illustration of the guaranteed accuracy of different randomized models with Gaussian (b) and Laplace (c) noises given the norm of the adversarial perturbation. The accuracies and entropies and estimated empirically.

rate of 0.1, 0.02, 0.004 and 0.00008 after respectively 7500, 15000 and 20000 steps. The networks achieve for CIFAR10 and 100 a TOP-1 accuracy of 95.8% and 79.1% respectively on test images. For ImageNet [11], we use an Inception ResNet v2 [41] which is the sate of the art architecture for this dataset and achieve a TOP-1 accuracy of 80%. For the training of ImageNet, we use the same hyper parameters setting as the original implementation. We train the network for 120 epochs with a batch size of 256, dropout 0.8 and Relu as activation function. All evaluations were done with a single crop on the non-blacklisted subset of the validation set.

To transform these classical networks to probabilistic mappings, we inject noise drawn from Laplace and Gaussian distributions, each with various standard deviations. While the noise could theoretically be injected anywhere in the network, we inject the noise on the image for simplicity. More experiments with noise injected in the first layer of the network are presented in the supplementary material. To evaluate our models under attack, we use three powerful iterative attacks with different norms: *ElasticNet* attack (EAD) [8] with $\ell_1$ distortion, *Carlini&Wagner* attack (C&W) [6] with $\ell_2$ distortion and *Projected Gradient Descent* attack (PGD) [27] with $\ell_\infty$ distortion. All standard deviations and attack intensities are in between $-1$ and $1$. Precise descriptions of our numerical experiments and of the attacks used for evaluation are deferred to the supplementary material.

**Attacks against randomized defenses:** It has been pointed out by [2, 5] that in a white box setting, an attacker with a complete knowledge of the system will know the distribution of the noise injected in the network. As such, to create a stronger adversarial example, the attacker can take the expectation of the loss or the logits of the randomized network during the computation of the attack. This technique is called Expectation Over Transformation (EoT) and we use a Monte Carlo method with 80 simulations to approximate the best perturbation for a randomized network.

## 5.2 Experimental results

**Trade-off between accuracy and intensity of noise (Q2.1):** When injecting noise as a defense mechanism, regardless of the distribution it is drawn from, we observe (as in Figure 1(a)) that the accuracy decreases when the noise intensity grows. In that sense, noise needs to be calibrated to preserve both accuracy and robustness against adversarial attacks, i.e. it needs to be large enough to preserve robustness and small enough to preserve accuracy. Figure 1(a) shows the loss of accuracy on CIFAR10 from $0.95$ to $0.82$ (respectively $0.95$ to $0.84$) with noise drawn from a Gaussian distribution (respectively Laplace) with a standard deviation from $0.01$ to $0.5$. Figure 1(b) and 1(c) illustrate the theoretical lower bound on accuracy under attack of Theorem 2 for different distributions and standard deviations. The term in entropy of Theorem 2 has been estimated using a Monte Carlo method with $10^4$ simulations. The trade-off between accuracy and robustness from Theorem 2 thus appears w.r.t the noise intensity. With small noises, the accuracy is high, but the guaranteed accuracy drops fast w.r.t the magnitude of the adversarial perturbation. Conversely, with bigger noises, the accuracy is lower but decreases slowly w.r.t the magnitude of the adversarial perturbation. These Figures also show that Theorem 2 gives strong accuracy guarantees against small adversarial perturbations. Next paragraph shows that in practice, randomized networks achieve much higher accuracy under attack than the theoretical bound, and against much larger perturbations.

Table 1: Accuracy under attack on the CIFAR-10 dataset with a randomized Wide ResNet architecture. We compare the accuracy on natural images and under attack with different noise over 3 iterative attacks (the number of steps is next to the name) made with 80 Monte Carlo simulations to compute EoT attacks. The first line is the baseline, no noise has been injected.

| Distribution | Sd | Natural | $\ell_1$ – EAD 60 | $\ell_2$ – C&W 60 | $\ell_\infty$ – PGD 20 |
|---|---|---|---|---|---|
| - | - | 0.958 | 0.035 | 0.034 | 0.384 |
| Normal | 0.01 | 0.954 | 0.193 | 0.294 | 0.408 |
|  | 0.50 | 0.824 | 0.448 | 0.523 | 0.587 |
| Laplace | 0.01 | 0.955 | 0.208 | 0.313 | 0.389 |
|  | 0.50 | 0.846 | 0.464 | 0.494 | 0.589 |

Table 2: Accuracy under attack of randomized neural network with different distributions and standard deviations versus adversarial training by Madry et al. [27]. The PGD attack has been made with 20 step, an epsilon of 0.06 and a step size of 0.006 (input space between $-1$ and $+1$). The Carlini&Wagner attack uses 30 steps, 9 binary search steps and a 0.01 learning rate. The first line refers to the baseline without attack.

| Attack | Steps | Madry et al. [27] | Normal 0.32 | Laplace 0.32 | Normal 0.5 | Laplace 0.5 |
|---|---|---|---|---|---|---|
| - | - | 0.873 | 0.876 | 0.891 | 0.824 | 0.846 |
| $\ell_\infty$ – PGD | 20 | 0.456 | 0.566 | 0.576 | 0.587 | 0.589 |
| $\ell_2$ – C&W | 30 | 0.468 | 0.512 | 0.502 | 0.489 | 0.479 |

**Performance of randomized networks under attacks and comparison to state of the art (Q2.2):**
While Figure 1(b) and 1(c) illustrated a theoretical robustness against growing adversarial perturbations, Table 1 illustrates this trade-off experimentally. It compares the accuracy under attack of a deterministic network with the one of randomized networks with Gaussian and Laplace noises both with low (0.01) and high (0.5) standard deviations. Randomized networks with a small noise lead to no loss in accuracy with a small robustness while high noises lead to a higher robustness at the expense of loss of accuracy ($\sim 11$ points). Table 2 compares the accuracy and the accuracy under attack of randomized networks with Gaussian and Laplace distributions for different standard deviations against adversarial training [27]. We observe that the accuracy on natural images of both noise injection methods are similar to the one from [27]. Moreover, both methods are more robust than adversarial training to PGD and C&W attacks. With all the experiments, to construct an EoT attack, we use 80 Monte Carlo simulations at every step the attacks. These experiments show that randomized defenses can be competitive given the intensity of noise injected in the network. Note that these experiments have been led with EoT of size 80. For much bigger sizes of EoT these results would be mitigated. Nevertheless, the accuracy would never drop under the bounds illustrated in Figure 5.2, since Theorem 2 gives a bound that on the worst case attack strategy (including EoT).

## 6 Conclusion and future work

This paper brings new contributions to the field of provable defenses to adversarial attacks. Principled answers have been provided to key questions on the interest of randomization techniques, and on their loss of accuracy under attack. The obtained bounds have been illustrated in practice by conducting thorough experiments on baseline datasets such as CIFAR and ImageNet. We show in particular that a simple method based on injecting noise drawn from the Exponential family is competitive compared to baseline approaches while leading to provable guarantees. Future work will focus on investigating other noise distributions belonging or not to the Exponential family, combining randomization with more sophisticated defenses and on devising new tight bounds on the adversarial risk gap.

**Acknowledgements:** This work was granted access to the OpenPOWER prototype from GENCI-IDRIS under the Preparatory Access AP010610510 made by GENCI. R. Pinot benefited from a JSPS Summer Program Fellowship during this work (Grant number SP18218). L. Meunier and J. Atif would also like to thank Adrien Balp from Société Générale for his support.

## Footnotes

[1]Name introduced in [9] which came later than [24].

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
