[Supplementary Material]

# Theoretical evidence for adversarial robustness through randomization
## - Supplementary Material -

## Contents

# 1  Notations and definitions

Let us consider an output space $\mathcal{Y}$, and $\mathcal{F}_{\mathcal{Y}}$ a $\sigma\text{-}algebra$ over $\mathcal{Y}$. We denote $\mathcal{P}(\mathcal{Y})$ the set of probability measures over $(\mathcal{Y}, \mathcal{F}_{\mathcal{Y}})$. Let $(\mathcal{Y}', \mathcal{F}_{\mathcal{Y}'})$ be a second measurable space, and $\phi$ a measurable mapping from $(\mathcal{Y}, \mathcal{F}_{\mathcal{Y}})$ to $(\mathcal{Y}', \mathcal{F}_{\mathcal{Y}'})$. Finally Let us consider $\mu, \nu$ two measures on $(\mathcal{Y}, \mathcal{F}_{\mathcal{Y}})$.

**Dominated measure:** $\mu$ is said to be dominated by $\nu$ (denoted $\mu \ll \nu$) if for all $Y \in \mathcal{F}_{\mathcal{Y}}$, $\nu(Y) = 0 \implies \mu(Y) = 0$. If $\mu$ is dominated by $\nu$, there is a measurable function $h : \mathcal{Y} \to [0, +\infty)$ such that for all $Y \in \mathcal{F}_{\mathcal{Y}}$, $\mu(Y) = \int_Y h\, d\nu$. $h$ is called the Radon-Nikodym derivative of $\mu$ w.r.t. $\nu$ and is denoted $\frac{d\mu}{d\nu}$.

**Push-forward measure:** the push-forward measure of $\nu$ by $\phi$ (denoted $\phi\#\nu$) is the measure on $(\mathcal{Y}', \mathcal{F}_{\mathcal{Y}'})$ such that $\forall Z \in \mathcal{F}_{\mathcal{Y}'}$, $\phi\#\nu(Z) = \nu(\phi^{-1}(Z))$.

**Convolution product:** the convolution of $\nu$ with $\mu$, denoted $\nu * \mu$ is the push-forward measure of $\nu \otimes \mu$ by the addition on $\mathcal{Y}$. Since the convolution between functions is defined accordingly, we use $*$ indifferently for measures and simple functions.

**Modulus of continuity:** Let us consider $f : (E, \|.\|_E) \to (F, \|.\|_F)$. $f$ admits a non-decreasing modulus of continuity regarding $\|.\|_E$ and $\|.\|_F$ if there exists a non-decreasing function $\omega_f^{E,F} : \mathbb{R}^+ \to \mathbb{R}^+$ such as for all $x, y \in E$, $\|f(y) - f(x)\|_F \leq \omega_f^{E,F}(\|x - y\|_E)$.

# 2  Main proofs

As a trade-off between completeness and conciseness, we delayed the proofs of our theorems to this section.

## 2.1  Proof of Lemma 1

**Lemma 1.** *Let* M *be a probabilistic mapping, and let $d_1$ and $d_2$ be two metrics on $\mathcal{P}(\mathcal{Y})$. If there exists a non decreasing function $\phi : \mathbb{R} \to \mathbb{R}$ such that $\forall \mu_1, \mu_2 \in \mathcal{P}(\mathcal{Y})$, $d_1(\mu_1, \mu_2) \leq \phi(d_2(\mu_1, \mu_2))$, then the following holds:*

$$\text{M is } d_2\text{-}(\alpha, \epsilon, \gamma)\text{-robust} \implies \text{M is } d_1\text{-}(\alpha, \phi(\epsilon), \gamma)\text{-robust}$$

*Proof.* Let us consider a probabilistic mapping M, $x \sim \mathcal{D}$, and $\tau \in B(\alpha)$, one has $d_1(\text{M}(x), \text{M}(x + \tau)) \leq \phi(d_2(\text{M}(x), \text{M}(x + \tau))) \leq \phi(\epsilon)$. Hence $\mathbb{P}_{x \sim \mathcal{D}}\left[\forall \tau \in B(\alpha), \, d_1(\text{M}(x + \tau), \text{M}(x)) \leq \phi(\epsilon)\right] \leq 1 - \gamma$. By inverting the inequality, one gets the expected result. $\qquad\square$

## 2.2  Proof of Proposition 1

**Proposition 1** (Renyi-robustness implies TV-robustness)**.** *Let* M *be a probabilistic mapping, then for all $\lambda \geq 1$:*

$$\text{M is } d_{R,\lambda}\text{-}(\alpha, \epsilon, \gamma)\text{-robust} \implies \text{M is } d_{TV}\text{-}(\alpha, \epsilon', \gamma)\text{-robust}$$

$$\text{with } \epsilon' = \min\left(\frac{3}{2}\left(\sqrt{1 + \frac{4\epsilon}{9}} - 1\right)^{1/2}, \frac{\exp(\epsilon + 1) - 1}{\exp(\epsilon + 1) + 1}\right).$$

*Proof.* Given two probability measures $\mu_1$ and $\mu_2$ on $(\mathcal{Y}, \mathcal{F}_{\mathcal{Y}})$, and $\lambda > 0$ one wants to find a bound on $d_{TV}(\mu_1, \mu_2)$ as a functional of $d_{R,\lambda}(\mu_1, \mu_2)$.

Thanks to [6], one has

$$d_{KL}(\mu_1, \mu_2) \geq 2d_{TV}(\mu_1, \mu_2)^2 + \frac{4d_{TV}(\mu_1, \mu_2)^4}{9}.$$

From which it follows that

$$d_{TV}(\mu_1, \mu_2)^2 \leq \frac{9}{4}\left(\sqrt{1 + \frac{4d_{KL}(\mu_1, \mu_2)}{9}} - 1\right)$$

One thus finally gets:

$$d_{TV}(\mu_1, \mu_2) \leq \frac{3}{2}\left(\sqrt{1 + \frac{4d_{KL}(\mu_1, \mu_2)}{9}} - 1\right)^{1/2}$$

Moreover, using inequality from [16], one gets:

$$d_{KL}(\mu_1, \mu_2) \geq \log\left(\frac{1 + d_{TV}(\mu_1, \mu_2)}{1 - d_{TV}(\mu_1, \mu_2)}\right) - \frac{2d_{TV}(\mu_1, \mu_2)}{1 + d_{TV}(\mu_1, \mu_2)}$$

For the sake of simplicity, since the second part of the right hand side of the equation is non increasing given $d_{TV}(\mu_1, \mu_2)$, and since $0 \leq d_{TV}(\mu_1, \mu_2) \leq 1$ one gets:

$$d_{KL}(\mu_1, \mu_2) + 1 \geq \log\left(\frac{1 + d_{TV}(\mu_1, \mu_2)}{1 - d_{TV}(\mu_1, \mu_2)}\right)$$

Hence, one gets:

$$\frac{\exp(d_{KL}(\mu_1, \mu_2) + 1) - 1}{\exp(d_{KL}(\mu_1, \mu_2) + 1) + 1} \geq d_{TV}(\mu_1, \mu_2)$$

By combining the two results, one obtains:

$$d_{TV}(\mu_1, \mu_2) \leq \min\left(\frac{3}{2}\left(\sqrt{1 + \frac{4d_{KL}(\mu_1, \mu_2)}{9}} - 1\right)^{1/2}, \frac{\exp(d_{KL}(\mu_1, \mu_2) + 1) - 1}{\exp(d_{KL}(\mu_1, \mu_2) + 1) + 1}\right).$$

To conclude for $\lambda > 1$ it suffices to use Lemma 1, and the monotony of Renyi divergence regarding $\lambda$. $\qquad\square$

## 2.3 Proof of Proposition 2

**Proposition 2** (Data processing inequality). *Let us consider a probabilistic mapping* $M : \mathcal{X} \to \mathcal{P}(\mathcal{Y})$. *Let us also denote* $\rho : \mathcal{Y} \to \mathcal{Y}'$ *a deterministic function. If* $U \sim M(x)$ *then the probability measure* $M'(x)$ *s.t.* $\rho(U) \sim M'(x)$ *defines a probabilistic mapping* $M' : \mathcal{X} \to \mathcal{P}(\mathcal{Y}')$.

*For any* $\lambda > 1$ *if* $M$ *is* $d_{R,\lambda}$-$(\alpha, \epsilon, \gamma)$ *robust then* $M'$ *is also is* $d_{R,\lambda}$-$(\alpha, \epsilon, \gamma)$ *robust.*

*Proof.* Let us consider $M$ a $d_{R,\lambda}$-$(\alpha, \epsilon, \gamma)$ robust algorithm. Let us also take $x \in \mathcal{X}$, and $\tau \in B(\alpha)$. Without loss of generality, we consider that $M(x)$, and $M(x + \tau)$ are dominated by the same measure $\mu$. Finally let us take $\rho$ a measurable mapping from $(\mathcal{Y}, \mathcal{F}_{\mathcal{Y}})$ to $(\mathcal{Y}', \mathcal{F}_{\mathcal{Y}'})$. For the sake of readability we denote $M(x) = \nu_1$ and $M(x + \tau) = \nu_2$ (therefore $M'(x) = \rho\#\nu_1$, and $M'(x + \tau) = \rho\#\nu_2$).

Since $\mu >> \nu_1, \nu_2$, one has $\rho\#\mu >> \rho\#\nu_1, \rho\#\nu_2$. Hence one has

$$d_{R,\lambda}(\rho\#\nu_1, \rho\#\nu_2) = \frac{1}{\lambda - 1} \log \int_{\mathcal{Y}'} \left(\frac{d\rho\#\nu_1}{d\rho\#\mu}\right)^\lambda \left(\frac{d\rho\#\nu_2}{d\rho\#\mu}\right)^{1-\lambda} d\rho\#\mu = \frac{1}{\lambda - 1} \log \int_{\mathcal{Y}'} \left(\frac{d\rho\#\nu_1}{d\rho\#\nu_2}\right)^\lambda d\rho\#\nu_2$$

Simply using the transfer theorem, one gets

$$d_{R,\lambda}(\rho\#\nu_1, \rho\#\nu_2) = \frac{1}{\lambda - 1} \log \int_{\mathcal{Y}} \left(\frac{d\rho\#\nu_1}{d\rho\#\nu_2} \circ \rho\right)^\lambda d\nu_2$$

Since $\left(\frac{d\rho\#\nu_1}{d\rho\#\nu_2} \circ \rho\right) = \mathbb{E}\left(\frac{d\nu_1}{d\nu_2}\big|\rho^{-1}(\mathcal{F}_{\mathcal{Y}'})\right)$ one easily gets the following:

$$d_{R,\lambda}(\rho\#\nu_1, \rho\#\nu_2) = \frac{1}{\lambda - 1} \log \int_{\mathcal{Y}} \left(\frac{d\rho\#\nu_1}{d\rho\#\nu_2} \circ \rho\right)^\lambda d\nu_2 = \frac{1}{\lambda - 1} \log \int_{\mathcal{Y}} \mathbb{E}\left(\frac{d\nu_1}{d\nu_2}\big|\rho^{-1}(\mathcal{F}_{\mathcal{Y}'})\right)^\lambda d\nu_2$$

Finally, by using the Jensen inequality, and the property of the conditional expectation, one has

$$d_{R,\lambda}(\rho\#\nu_1, \rho\#\nu_2) \leq \frac{1}{\lambda - 1} \log \int_{\mathcal{Y}} \mathbb{E}\left(\frac{d\nu_1}{d\nu_2}^\lambda\big|\rho^{-1}(\mathcal{F}_{\mathcal{Y}'})\right) d\nu_2 = \frac{1}{\lambda - 1} \log \int_{\mathcal{Y}} \frac{d\nu_1}{d\nu_2}^\lambda d\nu_2 = d_{R,\lambda}(\nu_1, \nu_2).$$

$\qquad\square$

### 2.4 Proof of Theorem 1

**Lemma 2.** *Let $\psi : \mathcal{X} \to \mathbb{R}^n$ be a mapping. For any $\alpha \geq 0$ and for any norms $||.||_A$ and $||.||_B$, one can define*
$\Delta_\alpha^{A,B}(\psi) := \sup\limits_{x,y \in \mathcal{X}, ||x-y||_A \leq \alpha} ||\psi(x)-\psi(y)||_B$. *Let $X$ be a random variable. We denote $\mathrm{M}(x)$ the probability measure of the random variable $\psi(x) + X$.*

- *If $X \sim E_F(\theta, t, k)$ where $t$ and $k$ have non-decreasing modulus of continuity $\omega_t$ and $\omega_k$.*

  *Then for any $\alpha \geq 0$, $\mathrm{M}$ defines a probabilistic mapping that is $d_{R,\lambda}$-$(\alpha, \epsilon)$ robust with $\epsilon = ||\theta||_2 \omega_t^{B,2}(\Delta_\alpha^{A,B}(\phi)) + \omega_k^{B,1}(\Delta_\alpha^{A,B}(\phi))$ where $||.||_2$ is the norm corresponding to the scalar product in the definition of the exponential family density function and $||.||_1$ is here the absolute value on $\mathbb{R}$.*

- *If $X$ is a centered Gaussian random variable with a non degenerated matrix parameter $\Sigma$. Then for any $\alpha \geq 0$, $\mathrm{M}$ defines a probabilistic mapping that is $d_{R,\lambda}$-$(\alpha, \epsilon)$ robust with $\epsilon = \frac{\lambda \Delta_\alpha^{A,2}(\phi)^2}{2\sigma_{min}(\Sigma)}$ where $||.||_2$ is the canonical Euclidean norm on $\mathbb{R}^n$.*

*Proof.* Let us consider $\mathrm{M}$ the probabilistic mapping constructed from noise injections respectively drawn from 1) an exponential family with non-decreasing modulus of continuity, or 2) a non degenerate Gaussian. Let us take $x \in \mathcal{X}$, and $\tau \in B(\alpha)$. Without loss of generality, we consider that $\mathrm{M}(x)$, and $\mathrm{M}(x + \tau)$ are dominated by the same measure $\mu$. Let us also denote, $p_F$ the Radon-Nikodym derivative of the noise drawn in 1) with respect to $\mu$, $p_G$ the Radon-Nikodym derivative of the noise drawn or in 2) with respect to $\mu$ and $\delta_a$ the Dirac function mapping any element to 1 if it equals $a$ and to 0 otherwise.

1)

$$
\begin{aligned}
d_{R,\lambda}\left(\mathrm{M}(x), \mathrm{M}(x+\tau)\right) &= d_{R,\lambda}\left(\nu * \delta_{\psi(x)}, \nu * \delta_{\psi(x+\tau)}\right) \\
&\leq d_{R,\infty}\left(\nu * \delta_{\psi(x)}, \nu * \delta_{\psi(x+\tau)}\right) \\
&= \log \sup_{z \in \mathbb{R}^n} \frac{(p_F * \delta_{\psi(x)})(z)}{(p_F * \delta_{\psi(x+\tau)})(z)} \\
&= \log \sup_{z \in \mathbb{R}^n} \exp(< t(z - \psi(x)) - t(z - \psi(x+\tau)), \theta > \\
&\quad + k(z - \psi(x)) - k(z - \psi(x+\tau))) \\
&\leq \sup_{z \in \mathbb{R}^n} ||\theta||_2 ||t(z - \psi(x)) - t(z - \psi(x+\tau))||_2 + |k(z - \psi(x)) - k(z - \psi(x+\tau))| \\
&\leq ||\theta||_2 \omega_t^{B,2}(||\psi(x+\tau) - \psi(x)||_B) + \omega_k^{B,1}(||\psi(x+\tau) - \psi(x)||_B) \\
&\leq ||\theta||_2 \omega_t^{B,2}(\Delta_\alpha^{A,B}(\psi)) + \omega_k^{B,1}(\Delta_\alpha^{A,B}(\psi))
\end{aligned}
$$

2)

$$
d_{R,\lambda}\left(\mathrm{M}(x), \mathrm{M}(x+\tau)\right) = \frac{1}{\lambda - 1} \log \int_{\mathbb{R}^n} \left(p_G * \delta_{\psi(x)}\right)^\lambda \times \left(p_G * \delta_{\psi(x+\tau)}\right)^{1-\lambda} d\mu
$$

$$
= \frac{1}{\lambda - 1} \log \int_{\mathbb{R}^n} \frac{\exp\left\{-1/2\left(\lambda \left(z - \psi(x)\right)^\top \Sigma^{-1} \left(z - \psi(x)\right) + (1 - \lambda)\left(z - \psi(x+\tau)\right)^\top \Sigma^{-1} \left(z - \psi(x+\tau)\right)\right)\right\}}{(2\pi)^{n/2}|\Sigma|^{1/2}} dz
$$

$$
= \frac{-\left(\lambda \psi(x)^\top \Sigma^{-1} \psi(x) + (1-\lambda)\psi(x+\tau)^\top \Sigma^{-1} \psi(x+\tau) - (\lambda \psi(x) + (1-\lambda)\psi(x+\tau))^\top \Sigma^{-1}\left(\lambda \psi(x) + (1-\lambda)\psi(x+\tau)\right)\right)}{2\lambda - 2}
$$

$$
= \frac{\lambda^2 - \lambda}{2(\lambda - 1)}(\psi(x) - \psi(x+\tau))^\top \Sigma^{-1}(\psi(x) - \psi(x+\tau))
$$

$$
\leq \frac{\lambda}{2}\sigma_{max}\left(\Sigma^{-1}\right)||(\psi(x) - \psi(x+\tau))||_2^2 \leq \frac{\lambda \Delta_\alpha^{A,2}(\psi)^2}{2\sigma_{min}(\Sigma)}.
$$

$\square$

**Theorem 1** (Exponential family ensures robustness). *Let us denote $\mathcal{N}_X^i(.) = \phi^n \circ ... \circ \phi^{i+1}(\mathcal{N}_{|i}(.) + X)$ with $X$ a random variable. Let us also consider $||.||_A$, and $||.||_B$ two arbitrary norms respectively on $\mathcal{X}$ and on the output space of $\mathcal{N}_X^i$.*

- *If $X \sim E_F(\theta, t, k)$ where $t$ and $k$ have non-decreasing modulus of continuity $\omega_t$ and $\omega_k$. Then for any $\alpha \geq 0$, $\mathcal{N}_X^i(.)$ defines a probabilistic mapping that is $d_{R,\lambda}$-$(\alpha, \epsilon)$ robust with $\epsilon = ||\theta||_2 \omega_t^{B,2}(\Delta_\alpha^{A,B}(\phi)) + \omega_k^{B,1}(\Delta_\alpha^{A,B}(\phi))$ where $||.||_2$ is the norm corresponding to the scalar product in the definition of the exponential family density function and $||.||_1$ is here the absolute value on $\mathbb{R}$. The notion of continuity modulus is defined in the preamble of this supplementary material.*

- *If $X$ is a centered Gaussian[1] random variable with a non degenerated matrix parameter $\Sigma$. Then for any $\alpha \geq 0$, $\mathcal{N}_X^i(.)$ defines a probabilistic mapping that is $d_{R,\lambda}$-$(\alpha, \epsilon)$ robust with $\epsilon = \frac{\lambda \Delta_\alpha^{A,2}(\phi)^2}{2\sigma_{min}(\Sigma)}$ where $||.||_2$ is the canonical Euclidean norm on $\mathbb{R}^n$.*

*Proof.* This theorem is a direct consequence of Lemma 2 and Proposition 2. By applying Lemma 2 to $\psi = \mathcal{N}_{|i}$ and Proposition 2 to $\rho = \phi^n \circ ... \circ \phi^{i+1}$, we immediately get the result. $\qquad\square$

## 2.5 Proof of Theorem 2

**Theorem 2** (Adversarial risk gap bound in the randomized setting). *Let $\mathrm{M}$ be the probabilistic mapping at hand. Let suppose that $\mathrm{M}$ is $d_{R,\lambda}$-$(\alpha, \epsilon)$ robust for some $\lambda \geq 1$ then:*

$$|\operatorname{Risk}_\alpha(\mathrm{M}) - \operatorname{Risk}(\mathrm{M})| \leq 1 - e^{-\epsilon}\mathbb{E}_x\left[e^{-H(\mathrm{M}(x))}\right]$$

*where $H$ is the Shannon entropy: $H(p) = -\sum_i p_i \log(p_i)$*

*Proof.* Let $\mathrm{M}$ be a randomized network with a noise $X$ injected at layer $i$. We have:

$$|\operatorname{Risk}_\alpha(\mathrm{M}) - \operatorname{Risk}(\mathrm{M})| = \left|\mathbb{E}_{(x,y)}\left[\sup_{\tau/||\tau||\leq\alpha}\mathbb{E}_{y'\sim\mathrm{M}(x+\tau)}\left[\mathbb{1}\left(y_1 \neq y\right)\right] - \mathbb{E}_{y_2\sim\mathrm{M}(x)}\left[\mathbb{1}\left(y' \neq y\right)\right]\right]\right|$$

$$= \left|\mathbb{E}_{(x,y)}\left[\sup_{\tau/||\tau||\leq\alpha}\mathbb{E}_{y_1\sim\mathrm{M}(x+\tau),y_2\sim\mathrm{M}(x)}\left[\mathbb{1}\left(y_1 \neq y\right) - \mathbb{1}\left(y_2 \neq y\right)\right]\right]\right|$$

$$\leq \mathbb{E}_{(x,y)}\left[\sup_{\tau/||\tau||\leq\alpha}\mathbb{E}_{y_1\sim\mathrm{M}(x+\tau),y_2\sim\mathrm{M}(x)}\left[|\mathbb{1}\left(y_1 \neq y\right) - \mathbb{1}\left(y_2 \neq y\right)|\right]\right]$$

$$\leq \mathbb{E}_{(x,y)}\left[\sup_{\tau/||\tau||\leq\alpha}\mathbb{E}_{y_1\sim\mathrm{M}(x+\tau),y_2\sim\mathrm{M}(x)}\left[\mathbb{1}\left(y_1 \neq y_2\right)\right]\right]$$

$$= \mathbb{E}_{(x,y)}\left[\sup_{\tau/||\tau||\leq\alpha}\mathbb{P}_{y_1\sim\mathrm{M}(x+\tau),y_2\sim\mathrm{M}(x)}\left(y_1 \neq y_2\right)\right]$$

For two discrete random independent variables of law $P = (p_1, ..., p_K)$ and $Q = (q_1, ..., q_K)$, thanks to Jensen's inequality:

$$\mathbb{P}(P = Q) = \sum_{i=1}^{K} p_i q_i \geq \exp\left(\sum_{i=1}^{K} p_i \log q_i\right) = \exp\left(-d_{KL}(P, Q) - H(P)\right)$$

Then we have:

$$\mathbb{E}_{(x,y)}\left[\sup_{\tau/||\tau||\leq\alpha}\mathbb{P}_{y_1\sim\mathrm{M}(x+\tau),y_2\sim\mathrm{M}(x)}\left(y_1 \neq y_2\right)\right] \leq \mathbb{E}_{(x,y)}\left[\sup_{\tau/||\tau||\leq\alpha} 1 - e^{-d_{KL}(\mathrm{M}(x),\mathrm{M}(x+\tau))-H(\mathrm{M}(x))}\right]$$

$$\leq \mathbb{E}_{(x,y)}\left[1 - e^{-\epsilon-H(\mathrm{M}(x))}\right]$$

$$= 1 - e^{-\epsilon}\mathbb{E}_x\left[e^{-H(\mathrm{M}(x))}\right]$$

$\qquad\square$

### 2.6 Proof of Theorem 3

**Theorem 3.** *Let $x \in \mathcal{X}$, and $\mathrm{M}$ be a probabilistic mapping with values in $\mathbb{R}^K$. If $\mathrm{M}$ is $d_{R,\lambda}$-$(\alpha, \epsilon)$ robust, and if there exist $k^*$ and $\delta^* \in (0,1)$ s.t. $\mathbb{E}_{y \sim \mathrm{M}(x)}[y_{k^*}] > e^{2\epsilon'} \max_{i \neq k^*} \mathbb{E}_{y \sim \mathrm{M}(x)}[y_i] + (1 + e^{\epsilon'})\delta^*$, with $\epsilon' = \epsilon + \frac{\log(1/\delta^*)}{\lambda - 1}$. Then, for the classifier $f : x \mapsto \underset{k \in [K]}{\mathrm{argmax}} \, \mathbb{E}_{y \sim \mathrm{M}(x)}[y_k]$ there is no perturbation $\tau \in \mathrm{B}(\alpha)$ such that $f(x) \neq f(x + \tau)$.*

*Proof.* Let $x \in \mathcal{X}$, and $\mathrm{M}$ a $d_\lambda$-$(\alpha, \epsilon)$ robust mapping.

1). By probability preservation [11], for any $Y \in \sigma(\mathcal{Y})$, $\tau \in \mathrm{B}(\alpha)$ one gets $\mathrm{M}(x)(Y) \leq (e^\epsilon \mathrm{M}(x + \tau)(Y))^{\frac{\lambda-1}{\lambda}}$. Let us now take $\delta \in (0,1)$. If $[e^\epsilon \mathrm{M}(x + \tau)(Y)]^{\frac{\lambda-1}{\lambda}} > \delta$, then $[e^\epsilon \mathrm{M}(x + \tau)(Y)]^{\frac{\lambda-1}{\lambda}} \leq e^\epsilon \mathrm{M}(x + \tau)(Y)\delta^{\frac{-1}{\lambda}} = e^{\epsilon'} \mathrm{M}(x)(Y)$. Else $[e^\epsilon \mathrm{M}(x + \tau)(Y)]^{\frac{\lambda-1}{\lambda}} \leq \delta$. Then for any $Y \in \sigma(\mathcal{Y})$, $\tau \in \mathrm{B}(\alpha)$, and $\delta \in (0,1)$, one has $\mathrm{M}(x)(Y) \leq e^{\epsilon'} \mathrm{M}(x + \tau)(Y) + \delta$.

2). Tanks to 1), for any $\tau \in \mathrm{B}(\alpha)$, and $\delta \in (0,1)$, one has $\mathbb{E}_{y \sim \mathrm{M}(x)}[y] \leq e^{\epsilon'} \mathbb{E}_{y \sim \mathrm{M}(x+\tau)}[y] + \delta$ (element wise). Hence, if there exists $k^*$, and $\delta^*$ such that $\mathbb{E}_{y \sim \mathrm{M}(x)}[y_{k^*}] > e^{2\epsilon'} \max_{i \neq k^*} \mathbb{E}_{y \sim \mathrm{M}(x)}[y_i] + (1 + e^{\epsilon'})\delta^*$, then for any $\tau \in \mathrm{B}(\alpha)$ one has $\mathbb{E}_{y \sim \mathrm{M}(x+\tau)}[y_{k^*}] > \dfrac{e^{2\epsilon'} \max_{i \neq k^*} \mathbb{E}_{y \sim \mathrm{M}(x+\tau)}[y_i] + (1 + e^{\epsilon'})\delta^* - \delta^*}{e^{\epsilon'}} > \max_{i \neq k^*} \mathbb{E}_{y \sim \mathrm{M}(x+\tau)}[y_i]$. This being true for any $\tau \in \mathrm{B}(\alpha)$, one gets the expected result. $\qquad\square$

## 3 Additional results and discussions

In this section, we give some additional results on both the strength of the Renyi-divergence and a bound on the risk gap for TV-distance.

### 3.1 About Renyi divergence

In the main submission, we chose to use the Renyi-robustness as the principled measure of robustness. Since Renyi-divergence is a good surrogate for the trivial distance (which is a risk of the $0 - 1$-loss for probabilistic mappings), we supported this statement by showing that Renyi-divergence is stronger than TV-distance. In this section, we extend this result to most of the classical divergences used in Machine Learning and show that Renyi-divergence is stronger than all of them.

Let us consider an output space $\mathcal{Y}$, $\mathcal{F}_\mathcal{Y}$ a $\sigma$-$algebra$ over $\mathcal{Y}$, and $\mu_1, \mu_2, \nu$ three measures on $(\mathcal{Y}, \mathcal{F}_\mathcal{Y})$, with $\mu_1, \mu_2$ in the set of probability measures over $(\mathcal{Y}, \mathcal{F}_\mathcal{Y})$ denoted $\mathcal{P}(\mathcal{Y})$. One has $\nu >> \mu_1, \mu_2$ and one denotes $g_1$ and $g_2$ the Radon-Nikodym derivatives with respect to $\nu$.

**The Separation distance:**

$$d_S(\mu_1, \mu_2) := \sup_{\{z\} \in \mathcal{F}_\mathcal{Y}} 1 - \frac{\mu_1(\{z\})}{\mu_2(\{z\})}.$$

**The Hellinger distance:**

$$d_H(\mu_1, \mu_2) := \left[ \int_\mathcal{Y} \left( \sqrt{g_1} - \sqrt{g_2} \right)^2 d\nu \right]^{1/2}.$$

**The Prokhorov metric:**

$$d_P(\mu_1, \mu_2) := \inf \left\{ \zeta > 0 : \mu_1(B) \leq \mu_2(B^\zeta) + \zeta \text{ for all Borel sets } B \right\} \text{ where } B^\zeta = \{x : \inf_{y \in B} d_{\mathcal{Y}'}(x, y) \leq \zeta\}.$$

**The Discrepancy metric:**

$$d_D(\mu_1, \mu_2) := \sup_{\text{all closed balls B}} |\mu_1(B) - \mu_2(B)|.$$

**Lemma 3.** *Given two probability measures $\mu_1$ and $\mu_2$ on $(\mathcal{Y}, \mathcal{F}_\mathcal{Y})$ the Separation metric and the Renyi divergence satisfy the following relation: $d_S(\mu_1, \mu_2) \leq d_{R,\infty}(\mu_1, \mu_2)$*

*Proof.* The function $x :\to 1 - x - |\ln(x)|$ is negative on $\mathbb{R}$, therefore for any $\{z\} \in \mathcal{Y}$ one has $1 - \frac{\mu_1(\{z\})}{\mu_2(\{z\})} \leq \left| \ln \frac{\mu_1(\{z\})}{\mu_2(\{z\})} \right|$, hence $\sup_{\{z\} \in \mathcal{F}_\mathcal{Y}} 1 - \frac{\mu_1(\{z\})}{\mu_2(\{z\})} \leq \sup_{\{z\} \in \mathcal{F}_\mathcal{Y}} \left| \ln \frac{\mu_1(\{z\})}{\mu_2(\{z\})} \right| \leq \sup_{Z \in \mathcal{F}_\mathcal{Y}} \left| \ln \frac{\mu_1(Z)}{\mu_2(Z)} \right| = d_{R,\infty}(\mu_1, \mu_2)$ $\qquad\square$

**Theorem 4.** *Let* M *be the probabilistic mapping, then for all* $\lambda > 1$ *if* M *is* $d_{R,\lambda}$-$(\alpha, \epsilon, \gamma)$-*robust the following assertions holds:*

(1) M *is* $d_H$-$(\alpha, \sqrt{\epsilon}, \gamma)$-*robust.*

(2) M *is* $d_P$-$(\alpha, \epsilon', \gamma)$-*robust **and*** $d_D$-$(\alpha, \epsilon', \gamma)$-*robust, for* $\epsilon' = \min\left( \frac{3}{2} \left( \sqrt{1 + \frac{4\epsilon}{9}} - 1 \right)^{1/2}, \frac{\exp(\epsilon+1)-1}{\exp(\epsilon+1)+1} \right).$

(3) M *is* $d_W$-$(\alpha, \epsilon', \gamma)$-*robust with* $\epsilon' = \min\left( \frac{3}{2diam(\mathcal{Y})} \left( \sqrt{1 + \frac{4\epsilon}{9}} - 1 \right)^{1/2}, \frac{\exp(\epsilon+1)-1}{diam(\mathcal{Y})(\exp(\epsilon+1)+1)} \right).$

(4) *if* $\lambda = \infty$, M *is* $d_S$-$(\alpha, \epsilon, \gamma)$-*robust.*

*Proof.*

(1) The proof is a simple adaptation of Proposition 1 using the inequality $d_H(\mu_1, \mu_2)^2 \leq d_{KL}(\mu_1, \mu_2)$ [5] and Lemma 1.

(2) Using the inequalities $d_D(\mu_1, \mu_2) \leq d_{TV}(\mu_1, \mu_2)$ [5] and $d_P(\mu_1, \mu_2) \leq d_{TV}(\mu_1, \mu_2)$ [8], the proof is immediate, using Theorem 1 and Lemma 1.

(3) The proof is adapted from Proposition 1 using the inequality $d_W(\mu_1, \mu_2) \leq diam(\mathcal{Y})d_{TV}(\mu_1, \mu_2)$ [5].

(4) The result is a straightforward application of Lemma 3, and Lemma 1.

$\square$

## 3.2 risk gap with TV-robustness

In the main paper, we give a bound on the risk gap based on the Renyi-robustness. We extend this result to the TV-robustness, highlighting the fact that risk gap could be derived from any of the above divergences.

**Theorem 5.** *Let* M *be the probabilistic mapping at hand. Let us suppose that* M *is* $d_{TV}$-$(\alpha, \epsilon)$ *robust then:*

$$| \operatorname{Risk}_\alpha(M) - \operatorname{Risk}(M)| \leq 1 - (\mathbb{E}_x\left[ e^{-H_c(M(x))} \right] - \epsilon)$$

*where* $H_c$ *is the collision entropy:* $H_c(p) = -\log(\sum_i p_i^2)$

*Proof.* For two discrete random independent variables of law $P = (p_1, ..., p_K)$ and $Q = (q_1, ..., q_K)$, thanks to Jensen's inequality:

$$\mathbb{P}(P = Q) = \sum_{i=1}^{K} p_i q_i = \sum_{i=1}^{K} p_i^2 - \sum p_i(p_i - q_i) \geq e^{-H_c(P)} - d_{TV}(P, Q)$$

because, for any $i \in [K]$, $p_i - q_i \leq d_{TV}(P, Q)$.

Then, the proof is a simple adaptation of the model of proof from Theorem 2. $\square$

# 4 Additional empirical evaluation

Due to space limitations, we had to defer the thorough description of our experimental setup and the results of some additional experiments.

## 4.1 Architectures & Hyper-parameters

We conduct experiments with 3 different dataset:

- CIFAR-10 and CIFAR-100 datasets, which are composed of 50K training samples, 10000 test samples and respectively 10 and 100 different classes. Images are trained and evaluated with a resolution of 32 by 32 pixels.
- ImageNet dataset, which is composed of $\sim 1.2M$ training examples, $50K$ test samples and 1000 classes. Images are trained and evaluated with a resolution of 299 by 299 pixels.

For CIFAR-10 and CIFAR-100 [9], we used a Wide ResNet architecture [17] which is a variant of the ResNet model from [7]. We used 28 layers with a widen factor of 10. We trained all the networks for 200 epochs, a batch size of 400, dropout 0.3 and Leaky Relu activation with a slope on $\mathbb{R}^-$ of 0.1. We used the cross entropy loss with Momentum 0.9 and a piecewise constant learning rate of 0.1, 0.02, 0.004 and 0.00008 after respectively 7500, 15000 and 20000 steps. The networks achieve for CIFAR10 and 100 a TOP-1 accuracy of 95.8% and 79.1% respectively on test images.

For ImageNet [4], we used an Inception ResNet v2 [14] which is the sate of the art architecture for this dataset and achieved a TOP-1 accuracy of 80%. For the training of ImageNet, we used the same hyper parameters setting as the original implementation. We trained the network for 120 epochs with a batch size of 256, dropout 0.8, Relu as activation function. All evaluations were done with a single crop on the non-blacklisted subset of the validation set.

### 4.2 Evaluation under attack

We evaluate our models against the strongest possible attacks from the literature using different norms ($\ell_1$, $\ell_2$ and $\ell_\infty$) which are all optimization based attacks. On their guide to evaluate robustness, Carlini et al. [2] proposed the three following attacks for each norm:

**$\ell_2$ – Carlini & Wagner attack and $\ell_1$ – ElasticNet attack**   The $\ell_2$ Carlini & Wagner attack ($C\&W$) introduced in [3] is formulated as:

$$\min_{x+r \in \mathcal{X}} c \times ||r||_2 + g(x+r)$$

where $g$ is a function such that $g(y) \geq 0$ iff $f(y) = l'$ with $l'$ the target class. The authors listed some $g$ functions. we choose the following one:

$$g(x) = \max(F_{k(x)}(x) - \max_{i \neq k(x)}(F_i(x)), -\kappa)$$

where $F$ is the softmax function and $\kappa$ a positive constant.

Instead of using box-constrained L-BFGS [15] as in the original attack, the authors use instead a new variable for $x+r$:

$$x + r = \frac{1}{2}(\tanh(w) + 1)$$

Then a binary search is performed to optimize the constant $c$ and ADAM or SGD for computing an optimal solution.

$\ell_1$ – ElasticNet attack is an adaptation of $\ell_2$ C&W attack where the objective is adaptive to $\ell_1$ perturbations:

$$\min_{x+r \in \mathcal{X}} c_1 \times ||r||_1 + c_2 \times ||r||_2 + g(x+r)$$

**$\ell_\infty$ – PGD attack.**   The PGD attack proposed by [12] is a risk of the iterative FGSM attack proposed in [10]. The goal of the adversary is to solve the following problem:

$$\operatorname*{argmax}_{||r||_p \leq \epsilon} \mathcal{L}(F_\theta(x+r), y)$$

In practice, the authors proposed an iterative method to compute a solution:

$$x^{t+1} = P_{x \oplus r}(x^t + \alpha \operatorname{sign}(\nabla_x \mathcal{L}(F_\theta(x^t), y)))$$

Where $x \oplus r$ is the Minkowski sum between $\{x\}$ and $\{r$ s.t. $||r||_p \leq \epsilon\}$, $\alpha$ a gradient step size, $P_S$ is the projection operator on $S$ and $x^0$ is randomly chosen in $x \oplus r$.

### 4.3 Detailed results on CIFAR-10 and CIFAR-100

Figure 2(a) presents the trade-off accuracy versus intensity of noise for the CIFAR-100 dataset. As for CIFAR-10, we observe that the accuracy decreases from 0.79 with a small noise (0.01) to ~0.55 with a higher noise (0.5). The Figures 2(b) and 2(c) are coherent with the theoretical guarantee of accuracy (Theorem 2) that the model can achieve under attack with a given perturbation and noise.

Table 1 and 2 summarize the results on the accuracy and accuracy under attack of CIFAR-10 and CIFAR-100 datasets with a Randomized Wide ResNet architecture given the standard deviation of the injected noise and the number of iterations of the attack. For PGD, we use an epsilon max of 0.06 and a step size of 0.006 for an input space of between -1 and +1. We show that injecting noise empirically helps defending neural networks against adversarial attacks.

Figure 1: (a) Impact of the standard deviation of the injected noise on accuracy in a randomized model on CIFAR-100 dataset with a Wide ResNet architecture. (b) and (c) illustration of the guaranteed accuracy of different randomized models with Gaussian (b) and Laplace (c) noises given the norm of the adversarial perturbation.

Table 1: Accuracy and Accuracy under attack of CIFAR-10 dataset

| | Natural | $\ell_1$ – EAD | | | $\ell_2$ – C&W | | | $\ell_\infty$ – PGD | | |
|---|---|---|---|---|---|---|---|---|---|---|
| | | 20 | 50 | 60 | 20 | 50 | 60 | 10 | 15 | 20 |
| **Normal (Sd)** | | | | | | | | | | |
| 0.010 | 0.954 | | 0.208 | 0.193 | 0.172 | 0.271 | 0.294 | 0.411 | 0.428 | 0.408 |
| 0.050 | 0.950 | 0.265 | 0.347 | 0.367 | 0.350 | 0.454 | 0.423 | 0.638 | 0.549 | 0.486 |
| 0.130 | 0.931 | 0.389 | 0.401 | 0.411 | 0.443 | 0.495 | 0.515 | 0.710 | 0.636 | 0.553 |
| 0.200 | 0.913 | 0.411 | 0.456 | | 0.470 | 0.481 | 0.516 | **0.724** | 0.629 | 0.539 |
| 0.320 | 0.876 | 0.442 | 0.450 | 0.445 | 0.475 | **0.522** | 0.499 | 0.720 | **0.641** | 0.566 |
| 0.500 | 0.824 | **0.453** | **0.513** | **0.448** | **0.503** | 0.494 | **0.523** | 0.694 | 0.608 | **0.587** |
| **Laplace (Sd)** | | | | | | | | | | |
| 0.010 | 0.955 | 0.167 | 0.190 | 0.208 | 0.184 | 0.279 | 0.313 | 0.474 | 0.423 | 0.389 |
| 0.050 | 0.950 | 0.326 | 0.315 | 0.355 | 0.387 | 0.458 | 0.448 | 0.630 | 0.534 | 0.515 |
| 0.130 | 0.929 | 0.388 | 0.426 | 0.435 | 0.461 | **0.515** | 0.493 | 0.688 | 0.599 | 0.538 |
| 0.200 | 0.919 | 0.417 | | **0.464** | 0.484 | 0.481 | 0.501 | 0.730 | 0.600 | 0.569 |
| 0.320 | 0.891 | **0.460** | 0.443 | 0.448 | 0.472 | 0.499 | **0.520** | **0.750** | **0.665** | 0.576 |
| 0.500 | 0.846 | 0.454 | **0.471** | **0.464** | **0.488** | | 0.494 | 0.721 | 0.650 | **0.589** |
| **Exponential (Sd)** | | | | | | | | | | |
| 0.010 | 0.953 | 0.153 | 0.174 | | 0.228 | 0.292 | 0.306 | 0.443 | 0.404 | 0.395 |
| 0.050 | 0.953 | 0.312 | 0.326 | 0.330 | 0.343 | 0.468 | 0.435 | 0.616 | 0.575 | 0.479 |
| 0.130 | 0.940 | 0.373 | 0.402 | 0.411 | 0.424 | | 0.504 | 0.679 | 0.585 | 0.526 |
| 0.200 | 0.936 | 0.394 | | 0.414 | 0.455 | **0.510** | 0.501 | 0.701 | 0.623 | 0.550 |
| 0.320 | 0.919 | **0.429** | 0.426 | 0.416 | **0.494** | 0.492 | 0.513 | 0.739 | 0.638 | 0.564 |
| 0.500 | 0.900 | 0.423 | **0.454** | **0.470** | 0.488 | 0.494 | **0.516** | **0.752** | **0.699** | **0.594** |

## 4.4 Large scale robustness

Adversarial training fails to generalize to higher dimensional datasets such as ImageNet. We conducted experiments with the large scale ImageNet dataset and compared our randomized neural network against large scale adversarial training proposed by Kurakin et al. [10]. One can observe from Table 3 that the model from Kurakin et al. is neither robust against recent $\ell_1$ nor $\ell_2$ iterative attacks such as EAD and C&W. Moreover, it offers a small robustness against $\ell_\infty$ PGD attack. Our randomized neural network with EoT attacks offers a small robustness on $\ell_1$ and $\ell_2$ attacks while being less robust against PGD.

## 4.5 Experiments with noise on the first activation

The aim of the following experiments is empirically illustrate the *Data processing inequality* in Proposition 2.

Table 4 and 5 present the experiments conducted with the same set of parameters as the previous ones on CIFAR-10 and CIFAR-100, but with the noise injected in the first activation layer instead of directly in the image. We observe from Table 4 that we can inject more noise with a marginal loss on accuracy. The accuracy under attack is presented in Table 5 for a selection of models.

Table 2: Accuracy and Accuracy under attack of CIFAR-100 dataset.

| | Natural | $\ell_1$ – EAD | | | $\ell_2$ – C&W | | | $\ell_\infty$ – PGD | | |
|---|---|---|---|---|---|---|---|---|---|---|
| | | 20 | 50 | 60 | 20 | 50 | 60 | 10 | 15 | 20 |
| **Normal (Sd)** | | | | | | | | | | |
| 0.010 | 0.790 | 0.235 | 0.234 | 0.228 | 0.235 | 0.318 | 0.316 | 0.257 | 0.176 | 0.187 |
| 0.050 | 0.768 | 0.321 | 0.294 | 0.320 | 0.357 | 0.377 | 0.410 | 0.377 | 0.296 | 0.254 |
| 0.130 | 0.726 | **0.357** | **0.371** | 0.349 | 0.387 | **0.427** | **0.428** | 0.414 | 0.319 | 0.260 |
| 0.200 | 0.689 | 0.338 | 0.350 | **0.384** | 0.394 | 0.381 | | 0.439 | 0.356 | 0.277 |
| 0.320 | 0.627 | 0.334 | 0.344 | 0.350 | 0.328 | 0.364 | 0.400 | **0.441** | 0.366 | 0.299 |
| 0.500 | 0.553 | 0.322 | 0.331 | 0.331 | 0.349 | 0.342 | 0.351 | 0.408 | **0.374** | **0.308** |
| **Laplace (Sd)** | | | | | | | | | | |
| 0.010 | 0.782 | 0.199 | 0.227 | 0.243 | 0.225 | 0.311 | 0.321 | 0.236 | 0.190 | 0.177 |
| 0.050 | 0.763 | 0.326 | 0.317 | 0.331 | 0.354 | 0.377 | 0.409 | 0.368 | 0.319 | 0.256 |
| 0.130 | 0.723 | 0.337 | 0.357 | 0.344 | **0.408** | 0.414 | 0.408 | 0.420 | 0.346 | 0.293 |
| 0.200 | 0.695 | **0.355** | 0.349 | **0.361** | 0.393 | 0.405 | 0.393 | 0.445 | 0.340 | 0.303 |
| 0.320 | 0.647 | 0.324 | **0.373** | 0.357 | 0.388 | 0.387 | 0.373 | **0.460** | 0.381 | 0.303 |
| 0.500 | 0.572 | 0.310 | 0.308 | 0.323 | 0.358 | 0.351 | 0.361 | 0.425 | **0.403** | **0.329** |
| **Exponential (Sd)** | | | | | | | | | | |
| 0.010 | 0.785 | 0.218 | 0.251 | 0.217 | 0.247 | 0.278 | 0.321 | 0.250 | 0.214 | 0.169 |
| 0.050 | 0.767 | 0.323 | 0.337 | 0.317 | 0.346 | 0.380 | 0.402 | 0.356 | 0.291 | 0.235 |
| 0.130 | 0.749 | 0.330 | | 0.356 | **0.403** | **0.444** | **0.421** | 0.400 | 0.328 | 0.266 |
| 0.200 | 0.731 | 0.345 | 0.361 | 0.357 | 0.388 | 0.424 | 0.406 | 0.427 | 0.340 | 0.267 |
| 0.320 | 0.703 | 0.349 | 0.351 | 0.340 | 0.388 | 0.439 | 0.399 | 0.433 | 0.351 | 0.280 |
| 0.500 | 0.673 | **0.387** | **0.378** | **0.378** | 0.396 | 0.435 | | **0.485** | **0.370** | **0.322** |

Table 3: Accuracy under attack of the Adversarial model training by Kurakin et al. [10] and an Inception Resnet v2 model training with normal 0.1 noise injected in the image on the ImageNet dataset.

| | Baseline | $\ell_1$ EAD 60 | $\ell_2$ C&W 60 | $\ell_\infty$ PGD |
|---|---|---|---|---|
| **Kurakin et al. [10]** | 0.739 | 0.097 | 0.100 | 0.239 |
| **Normal 0.1** | 0.625 | 0.255 | 0.301 | 0.061 |

Table 4: Impact of the distribution and the intensity of the noise with randomized networks with noise injected on the first activation

| Sd | Normal | Sd | Laplace | Sd | Exponential |
|---|---|---|---|---|---|
| 0.01 | 0.956 | 0.01 | 0.955 | 0.01 | 0.953 |
| 0.23 | 0.943 | 0.05 | 0.947 | 0.08 | 0.943 |
| 0.45 | 0.935 | 0.10 | 0.933 | 0.15 | 0.938 |
| 0.68 | 0.926 | 0.15 | 0.916 | 0.23 | 0.925 |
| 0.90 | 0.916 | 0.20 | 0.911 | 0.30 | 0.919 |
| 1.00 | 0.916 | 0.25 | 0.897 | 0.38 | 0.903 |
| 1.34 | 0.906 | 0.30 | 0.889 | 0.45 | 0.897 |
| 1.55 | 0.900 | 0.35 | 0.882 | 0.53 | 0.886 |
| 1.77 | 0.893 | 0.40 | 0.867 | 0.60 | 0.885 |
| 2.00 | 0.886 | 0.45 | 0.855 | 0.68 | 0.875 |

Table 5: Accuracy and Accuracy under attack of selected models with noise on the first activation

| Dataset | Distribution | Sd | Natural | $\ell_1$ – EAD | | | $\ell_2$ – C&W | | | $\ell_\infty$ – PGD | |
|---|---|---|---|---|---|---|---|---|---|---|---|
| | | | | 20 | 50 | 60 | 20 | 50 | 60 | 10 | 20 |
| CIFAR10 | Normal | 1.55 | 0.900 | 0.441 | 0.440 | 0.413 | 0.477 | 0.482 | 0.484 | 0.683 | 0.526 |
| | Laplace | 0.25 | 0.897 | 0.388 | 0.436 | **0.445** | 0.481 | 0.506 | 0.491 | 0.664 | 0.493 |
| | Exponential | 0.38 | **0.903** | **0.456** | **0.463** | 0.438 | **0.495** | 0.516 | 0.506 | 0.697 | 0.557 |
| CIFAR100 | Normal | 0.45 | 0.741 | **0.362** | 0.352 | 0.353 | 0.352 | 0.410 | 0.418 | 0.380 | 0.250 |
| | Laplace | 0.10 | **0.742** | 0.350 | **0.367** | 0.350 | 0.371 | **0.419** | | 0.418 | **0.264** |
| | Exponential | 0.15 | 0.741 | 0.354 | 0.356 | **0.373** | **0.394** | 0.409 | **0.420** | **0.430** | 0.258 |

# 5 Additional discussions on the experiments

For the sake of completeness and reproducibility, we give some additional insights on the noise injection scheme and comprehensive details on our numerical experiments.

## 5.1 On the need for injecting noise in the training phase

Robustness has always been thought as a property to be enforced at inference time and it is tempting to focus only on injecting noise at inference. However, simply doing so ruins the accuracy of the algorithm (as it becomes an instance of distribution shift [13]). Indeed, making the assumption that the training and test distributions matches, in practice, injecting some noise at inference would result in changing the test distribution.

Distribution shift occurs when the training distribution differs from the test distribution. This implies that the hypothesis minimizing the empirical risk is not consistent, i.e. it does not converge to the true model as the training size increases. A way to circumvent that is to ensure that training and test distributions matches using importance weighting (in the case of covariate-shift) or with noise injection in the training phases as well (in our case).

## 5.2 Reproducibility of the experiments

We emphasize that all experiments should be easily reproducible. All our experiments are developed with TensorFlow version 1.12 [1]. The code is available on github:

```
https://github.com/MILES-PSL/Adversarial-Robustness-Through-Randomization.
```

The repository contains a *readme* file containing a small documentation on how to run the experiments, a configuration file which defines the architecture and the hyper-parameters of the experiments, python scripts which generate a bash command to run the experiments. The code contains Randomized Wide Resnet used for CIFAR-10 and CIFAR100, Inception Resnet v2 used for ImageNet, PGD, EAD and C&W attacks used for evaluation under attack. We ran our experiments, on a cluster with computers each having 8 GPU Nvidia V100.

## Footnotes

[1]Although the Gaussian distribution belongs to the exponential family, it does not satisfy the modulus of continuity constraint on $t$ and its robustness has to be proved differently.