[Reviews · NeurIPS 2019]

Reviewer 1



-- The two theoretical results are, to the best of my knowledge, novel and make use of well-established ideas from information and probability theory in an interesting manner for their proofs. My concern with both the results is their significance and interpretation. The first theorem claims to show that the use of the exponential family guarantees robustness, but the actual content of the theorem concerns the growth of the divergence with the parameters of the added noise. The probability of this divergence being large is what actually controls the risk. The relationship between the noise parameters and this growth is not elucidated in the paper, making it hard to see how the robustness actually comes about theoretically. Further, the theorem concerns a quantity that has been derived from the true risk on line 99, but the connection between misclassification (the true risk) and the magnitude of the divergence used is not clear. -- For Theorem 2, from the plots in Figure 1, the accuracy guarantee falls to 0 even with a small adversarial budget, especially for smaller noise magnitudes. It would be instructive to provide a discussion of how the accuracy guarantee can be tightened since, for real classifiers, the accuracy is much higher. Further, the gap in Theorem 2 is between two true risks, so referring to it as ‘generalization’, which is a term usually used for the gap in performance during training and test, is a misnomer. -- The evaluation results are interesting but need more clarity in their presentation. For example, in Figure 1, how is the accuracy guarantee with no adversary calculated when from Theorem 2, only the gap can be computed? Is that an empirically obtained quantity? If yes, then that has to be mentioned. -- In general, related work has been cited adequately but a reference to Ford et al. (Adversarial Examples are a Natural Consequence of Test Error in Noise) is missing, in spite of it being very closely related.

Reviewer 2



Many adversarial defense techniques have been developed recently for improving the robustness of neural networks against different adversarial attacks. This paper focuses on defending neural networks by injecting random noise drawn from a subclass of the exponential family of distributions during both training and testing. As listed above, the paper makes significant novel contributions to formalizing robustness in the considered setting and provides both experimental and theoretical evidence for increased robustness. However, I have a few concerns below I hope the authors can address during rebuttal: 1. Line 72-74 states that the paper provides results on certified robustness. However, the setting considered in this paper is probabilistic and does not provide absolute guarantees on the absence of "any" adversarial example in an adversarial region. There has been considerable work on training with certified defenses, see [1][2] and [3]. It would be good if the authors can compare and separate their work from those based on certified defenses. 2. In table 2, it is not clear what conclusion can be drawn from the reduced success rates for the two attacks wrt Madry et al. 2018. Can it be the case that the addition of noise make the attack problem harder for both attacks and thus they are unable to find adversarial examples while the number of adversarial examples remains the same as Madry et al. 2018? References: [1] Provable defenses against adversarial examples via the convex outer adversarial polytope. ICML'18 [2]Differentiable Abstract Interpretation for Provably Robust Neural Networks, ICML 2018. [3] On the Effectiveness of Interval Bound Propagation for Training Verifiably Robust Models, Arxiv 2018.

Reviewer 3



The paper considers adversarial robustness for models with random noises added, and provides some theoretical insights to understand the improvement in robustness under this setting. The main theorem in this paper basically says that: if the output of a deterministic function f(.) can be bounded by \Delta when the input is perturbed within a ball of radius \alpha, the Renyi divergence (a generalized K-L divergence) of the output of the network, plus some exponential family noise, are epsilon-close even under perturbation. In the Gaussian noise setting, robustness is in proportion to the minimum singular value of the covariance matrix of Gaussian distribution, which makes sense. The authors also conduct experiments using different noise distributions in exponential family. Several questions: 1. The authors defined d_p(y)-(\alpha,\epsilon,\gamma) robustness in a probabilistic manner. However it seems the main theorem actually gives a deterministic bound (the probability \gamma = 0). Is it possible to improve the main theorem to a probabilistic version, and uncover the relationship between gamma and epsilon? 2. The theoretical lower bound on accuracy under attack of Theorem 2 needs the entropy term. It can be estimated using a Monte Carlo method however there is no guarantee that the sampled entropy is close to the true entropy, thus there is no real guarantee here. The derived bound thus cannot give any provable robustness certificate for a given network. Overall, this is a good paper that proposes something promising to explain the robustness improvements from randomization. I recommend to accept this paper, however the authors should make sure to address the issues above. *************** After reading the rebuttal, my first question has been addressed. However I still have concerns with the sampling method to obtain the entropy - although theoretically it is asymptotically converging but it might need a large amount of examples, especially the experiments were done using only a few samples. Overall, I am okay to recommend accepting the paper, but this paper certainly has some limitations and still has room for improvement.

[Author Response · NeurIPS 2019]

Our gratitude to the reviewers for their constructive comments and the useful references they pointed out. We will
revise the paper accordingly.

**Takeaway message from Theorem 1.** Following Rev.1's comments on the significance and interpretation of Theorem
1, we will clarify the message in the final version of the paper. Theorem 1 states that noise injection ensures robustness
(in the sense of Def 2). The degree of robustness, $\epsilon$ in Theorem 1, is in $O(||\theta||)$, where $\theta$ is the parameter of a given
Exponential distribution $E_F$. Since $\theta$ is decreasing w.r.t. $std(E_F)$ in general, the larger the amount of the added noise
is, closer to each other are the output distributions of the randomized classifier. The control of the added noise and its
impact on the accuracy gap is the subject of Theorem 2 (see discussion at lines 219-228).

**About the notion of certified accuracy.** Reviewers discussed our use of the term "certified accuracy". What we named
"certified accuracy" was indeed a probabilistic guarantee (this will be precised in the paper), and thus is different from
the notion mentioned in papers reviewers referred to. Giving this kind of "certified accuracy" bounds was not the main
focus of our work, nevertheless, we are able to devise certificates in the spirit of these papers by leveraging our concepts.
Notably, our work covers the certificate obtained in [LAG$^+$18] as presented below.

**Theorem** *Let $x \in \mathcal{X}$ be some input vector, and $\mathrm{M}$ be a probabilistic mapping such that for any $y \sim \mathrm{M}(x)$, $y =$*
*$(y_i)_{i \in [K]}$ is a probability vector of size $K$. If $\mathrm{M}$ is $d_\lambda$-$(\alpha, \epsilon)$ robust, and if there is some $k^*$, and some $0 < \delta^* < 1$*
*for which $\mathbb{E}_{y \sim \mathrm{M}(x)}[y_{k^*}] > e^{2\epsilon'} \max_{i \neq k^*} \mathbb{E}_{y \sim \mathrm{M}(x)}[y_i] + (1 + e^{\epsilon'})\delta^*$, with $\epsilon' = \epsilon + \frac{\log(1/\delta^*)}{\lambda - 1}$. Then, for the classifier*
*$f(.) = \operatorname{argmax}_k \mathbb{E}_{y \sim \mathrm{M}(.)}[y_k]$ there is no perturbation $\tau \in \mathrm{B}(\alpha)$ such that $f(x) \neq f(x + \tau)$.*

**Proof** *Let us consider some $x \in \mathcal{X}$. If $\mathrm{M}$ is $d_\lambda$-$(\alpha, \epsilon)$ robust, then with a proof similar to [Mir17] Proposition 3,*
*one easily gets that $\mathbb{E}_{y \sim \mathrm{M}(x)}[y] \leq e^{\epsilon + \frac{\log(1/\delta)}{\lambda - 1}} \mathbb{E}_{y \sim \mathrm{M}(x+\tau)}[y] + \delta$ (element wise), for any $\delta \in (0, 1)$. Then one can*
*use [LAG$^+$18] Proposition 1 to get the expected result.*

**On the probabilistic extensions of the theorems.** Rev.3 pointed out that the results we show in Theorem 1 only hold
for $\gamma = 0$. It is possible to extend the result for $\gamma > 0$. To do so, one possible workaround is to replace the robustness
guarantee in Theorem 1 as follows: adding noise drawn from $E_F(\theta, t, k)$ ensures that the randomized network is
$d_{R,\lambda}$-$(\alpha, \epsilon, \gamma)$ robust with $\epsilon = ||\theta||_2 \omega_t^{B,2}(L) + \omega_k^{B,1}(L)$ and $\gamma = \mathbb{P}_{x \sim \mathcal{D}_x}(\exists \tau \in B_A(\alpha), ||\phi(x + \tau) - \phi(x)||_B > L)$.
The same guarantee can be derived for the Gaussian case. Theorem 2 can also be extended: if $\mathrm{M}$ is $d_{R,\lambda}$-$(\alpha, \epsilon, \gamma)$ robust
for some $\lambda \geq 1$ then: $|\operatorname{Risk}_\alpha(\mathrm{M}) - \operatorname{Risk}(\mathrm{M})| \leq 1 - (1 - \gamma)e^{-\epsilon}\mathbb{E}_x\left[e^{-H(\mathrm{M}(x))}\right]$. However, in practice, it is intractable
to compute $\gamma$ as the data distribution is unknown. A discussion will be added in the paper.

**On the convergence of the entropy estimator.** Rev.3 raised a question about the convergence of the entropy estimator.
We used the MLE estimator from [Pan03] which is endowed with convergences guarantees. By integrating the correcting
bias (Miller-Madow estimator, Section 3), we can derive bounds for the entropy estimator. We will clarify this point in
the final version of the paper.

**Accuracy guarantee for the case with no adversary.** Studying the impact of corrupted noise on generalization is
another line of research, complementary to the scope of this paper. Outside some specific cases (GLM, Gaussian noise,
etc.), proving generalization bounds w.r.t. noise injection in general settings is still an open question to the best of our
knowledge.

**Other remarks**.
• *Misnomer of "generalization":* We agree with Rev.1. In the final version we will use "risk gap" instead.
• *On Table 1:* The accuracy without attack is indeed evaluated empirically. We will clarify that in the final version.
• *On Table 2:* We share the same intuition with Rev.2: the noise makes the attack problem harder for both attacks.
• *PGD iterations:* For the sake of comparison we used the same number of iterations as in [MMS$^+$18] (Table 2). We
are eager to use experiments with a large number of iterations and report the results in the final version of the paper.

# References

[LAG$^+$18] M. Lecuyer, V. Atlidakis, R. Geambasu, D. Hsu, and S. Jana. Certified robustness to adversarial examples
with differential privacy. In *2019 IEEE Symposium on Security and Privacy (SP)*, pages 727–743, 2018.

[Mir17] Ilya Mironov. Renyi differential privacy. In *IEEE Computer Security Foundations Symposium*, 2017.

[MMS$^+$18] Aleksander Madry, Aleksandar Makelov, Ludwig Schmidt, Dimitris Tsipras, and Adrian Vladu. Towards
deep learning models resistant to adversarial attacks. In *ICLR*, 2018.

[Pan03] Liam Paninski. Estimation of entropy and mutual information. *Neural computation*, 2003.


[Meta-Review · NeurIPS 2019]

The paper makes a contribution in theoretical analysis of randomization as a defence against model evasion attacks. All reviewers view the submission weakly positively, but note a number of potential improvements. Despite these minor weaknesses, the contribution appears useful enough contribution to warrant acceptance. For the final version, I would strongly urge the authors to consider the suggestions given by the reviewers. It is especially important to make absolutely clear the distinction between certified guarantees and the probabilistic guarantees obtained here, and avoid using terminology that might confuse these.